# Bandwidth-Aware Rescheduling Mechanism in SDN-Based Data Center Networks

**Ming-Chin Chuang [1,\*], Chia-Cheng Yen [2] and Chia-Jui Hung [1]**

1   Department of Computer Science and Information Engineering, China University of Technology, Taipei 11695, Taiwan; davidhung419@gmail.com
2   Department of Computer Science, University of California, Davis, CA 95616, USA; ccyen@ucdavis.edu
\*   Correspondence: speedboy@cute.edu.tw

**Abstract:** Recently, with the increase in network bandwidth, various cloud computing applications have become popular. A large number of network data packets will be generated in such a network. However, most existing network architectures cannot effectively handle big data, thereby necessitating an efficient mechanism to reduce task completion time when large amounts of data are processed in data center networks. Unfortunately, achieving the minimum task completion time in the Hadoop system is an NP-complete problem. Although many studies have proposed schemes for improving network performance, they have shortcomings that degrade their performance. For this reason, in this study, we propose a centralized solution, called the bandwidth-aware rescheduling (BARE) mechanism for software-defined network (SDN)-based data center networks. BARE improves network performance by employing a prefetching mechanism and a centralized network monitor to collect global information, sorting out the locality data process, splitting tasks, and executing a rescheduling mechanism with a scheduler to reduce task completion time. Finally, we used simulations to demonstrate our scheme's effectiveness. Simulation results show that our scheme outperforms other existing schemes in terms of task completion time and the ratio of data locality.

**Keywords:** software-defined network; rescheduling; bandwidth-aware; Hadoop

## 1. Introduction

Recently, several cloud application services [1,2] have been proposed due to an increase in network bandwidth. Many conceptual applications, such as cloud-based artificial intelligence services, remote patient–doctor telemedicine, real-time mixed reality, and big data collection and analysis in industrial Internet-of-Things applications, will be realized in the near future. In this case, large amounts of data are generated and transmitted through networks. Many well-known cloud service providers (e.g., Google, Microsoft, Amazon, Facebook, and Salesforce) have deployed multiple data centers in multiple locations to provide customers with quick access to their services and improve user experience.

In order to process data efficiently and concurrently on a large scale, an open-source Hadoop [3] is proposed. Hadoop provides a software framework for distributed storage and processing of big data using the MapReduce model [4]. One of the challenges in cloud computing is developing an efficient job management scheme in data centers. A job is referred to as a request, and local or remote nodes are required to complete the job and return a result. As the number of job requests increases in the future, the computation load of data centers will also increase. Although many studies [5–7] have proposed schemes for improving the performance of Hadoop, these schemes have shortcomings that degrade their performance. Therefore, this study aims to find an effective task scheduling method that overcomes the shortcomings of previous methods, further reducing task completion time (TCT).

In addition, several research results [8,9] have confirmed that software-defined networks (SDNs) [10] can significantly improve the performance of data centers. In reality, SDNs have also been widely introduced and adapted in the industry. Therefore, our solution, known as the bandwidth-aware rescheduling mechanism (BARE), is also based on SDNs. The core idea of the algorithm is to maximize efficiency by comparing remote task execution time and data migration time, thereby improving data locality and reducing task execution time. The main contributions of this study are summarized as follows:

- We formulate a scheduling problem for different computation times of tasks for each node and propose a task splitting and rescheduling mechanism to reduce the TCT and raise the ratio of data locality (RDL);
- The proposed algorithm uses a prefetching mechanism to further improve the overall completion time;
- We performed several simulations to evaluate the algorithm's efficiency. Simulation results show that the proposed scheme has better performance in terms of TCT and data locality ratio.

The remainder of this paper is organized as follows: Section 2 provides a review of related work. Section 3 describes problem formulation. Section 4 describes the proposed scheduling scheme, and Section 5 shows the algorithm's performance results. Section 6 contains some concluding remarks.

## 2. Related Work

### 2.1. Software-Defined Network (SDN)

An SDN is an emerging network framework that separates the control plane from the data plane. There are three layers in an SDN architecture: the application, control, and infrastructure layers. The application layer uses application programming interfaces to communicate with controllers. The OpenFlow protocol [11] is the first standardized protocol defined between the control and infrastructure layers, allowing network administrators to determine how data flows should be routed between switches and network entities in networks. With flexibility and central management characteristics, wireless networks can benefit from SDN evolution to meet the burgeoning 5G capacity. Studies [12–16] surveyed several kinds of challenges and issues in SDNs. Figure 1 shows the SDN architecture.

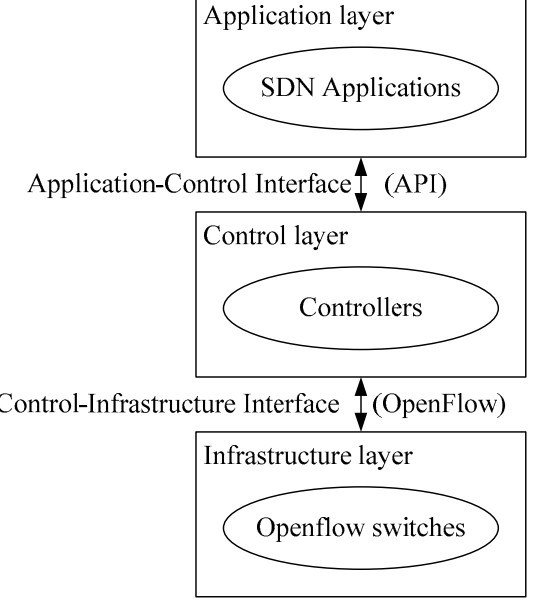

**Figure 1.** SDN (software-defined network) architecture.

### 2.2. Hadoop Default Scheduler (HDS)

In many cases, the Hadoop system is introduced to handle job scheduling and node assignments. The goal of the Hadoop default scheduler (HDS) [17] is to find an idle node and assign jobs to it. In Hadoop, the default scheduler assigns jobs to nodes on a first-in, first-out (FIFO) basis. However, the job size and the capability of nodes are not considered. Therefore, processing delays due to job size and the capability of nodes are expected and have been evaluated [18]. The HDS randomly compares the execution time of two nodes and then determines the faster node for tasks. The flowchart of the HDS is shown in Figure 2. In Figure 3, we assume that the HDS splits a job equally into nine tasks (TK), taking 9 s to process each task and transfer their replicas randomly to four nodes for processing data. The movement time of each task is 5 s. Since each node has a previous job to complete, the initial time varies. Note that during the comparison of the nodes' completion time, Node 4 has the least amount of time for TK9. If Node 4 does not have a replica of TK9, the scheduler moves TK9 to Node 4 to process the task. In this example, the total completion time is 39 s. The HDS does not consider that the transmission time of a file leads to nonoptimal performance.

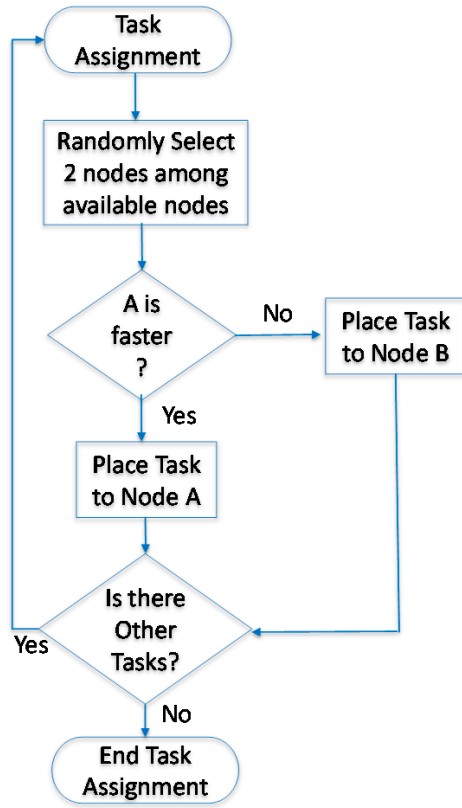

**Figure 2.** Flowchart of the Hadoop default scheduler.

### 2.3. Balance-Reduce Scheduler

In order to improve the effectiveness of the HDS, a balance-reduce scheduler (BAR) scheme [5] is proposed. The scheduler is composed of job and task trackers. Leveraging the Hadoop and centralized scheduler, the BAR method globally adjusts the tasks and node assignments of a job. As a result, a job is divided into multiple tasks, and the scheduler effectively assigns them to available nodes. Figure 4 shows the flowchart of the BAR, which is a two-phase method. The first phase is similar to the HDS method, i.e., it divides and assigns tasks to each node. In the second phase, the BAR method searches and evaluates the time of the current nodes and assigns a task with the longest time length to an idle node with the least time, thereby reducing the total process time. Following the same example as

the HDS, the BAR method considers the time of moving a task from one node to another. It then keeps TK9 at its original node in the process. Although there is no movement between nodes, the overall completion time is reduced to 38 s, as shown in Figure 5.

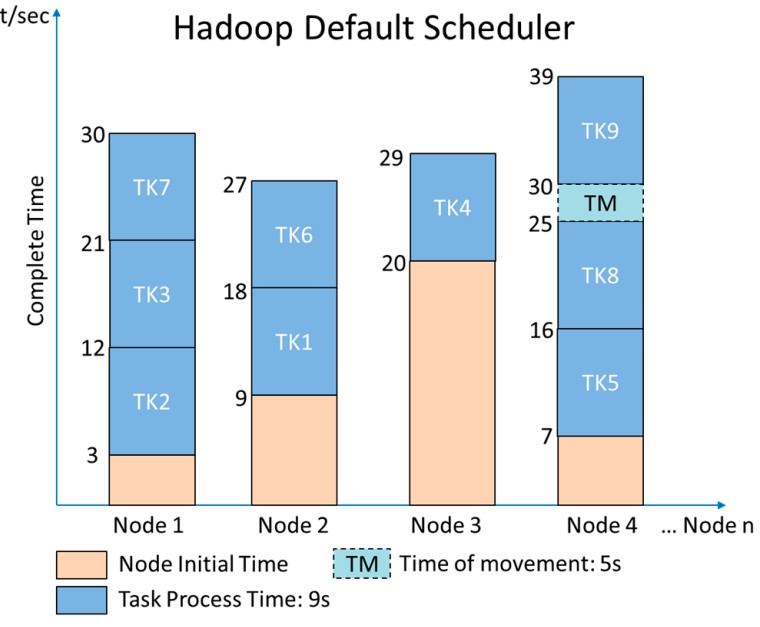

**Figure 3.** An example of HDS task assignments.

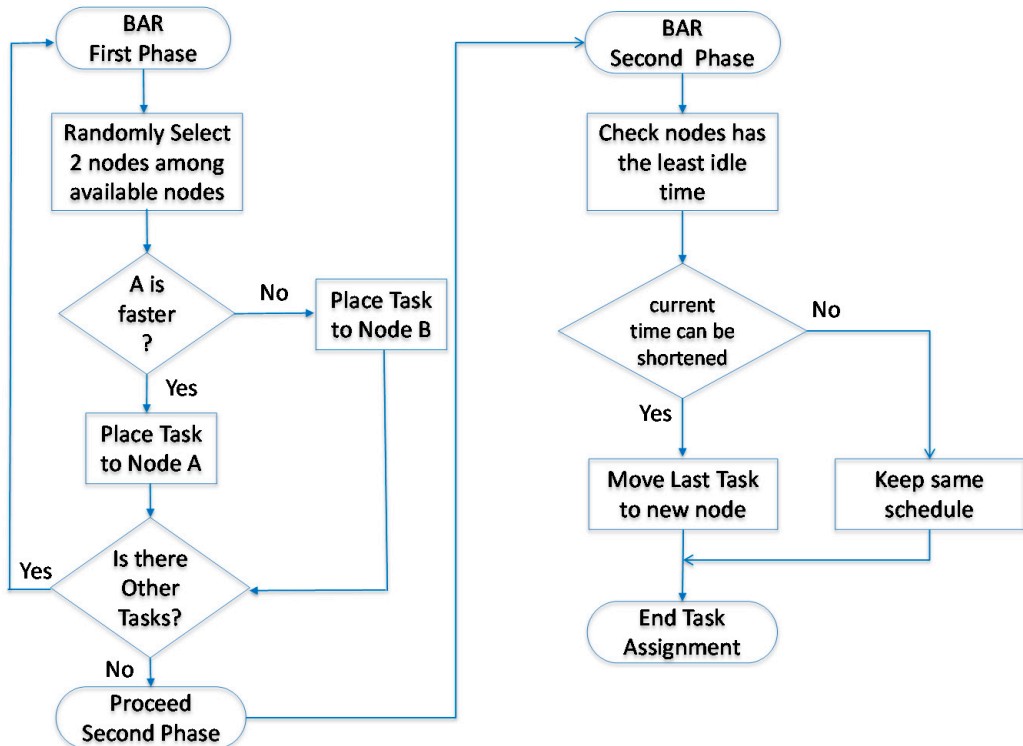

**Figure 4.** Flowchart of BAR (balance-reduce scheduler).

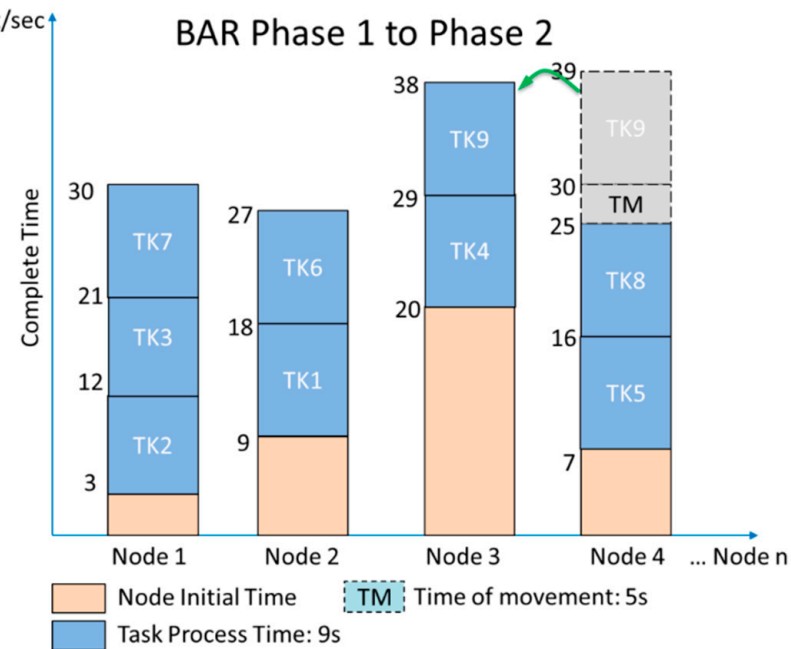

**Figure 5.** An example of BAR task assignments.

*2.4. Bandwidth-Aware Scheduling with SDN in Hadoop*

Bandwidth-aware scheduling with SDNs in Hadoop (BASS) [6,7] is a method based on the BAR and Hadoop. BASS treats task efficiency as an NP-complete problem [19–21] which defines, schedules, and evaluates the cost of tasks and then simulates tasks to an optimal solution after complex computation. In addition, the BASS method considers the bandwidth of each node, and tasks are assigned to nodes that have the bandwidth to handle such tasks. Then, the Pre-BASS scheme, which adds a prefetching scheme based on BASS, is proposed. Figure 6 shows the flowchart of BASS and Pre-BASS, and Figure 7 shows an example of task assignments by BASS. TK1 only has a replica on Node 2 and Node 3, and their process completion times are 18 and 29 s, respectively. From a global perspective, Node 1 has the least completion time, including movement time for the task. The total completion time of TK1 is 17 s; this means that Node 1 is the best choice among the nodes. Therefore, the overall completion time for the BASS method is only 35 s.

With another method [7], leveraging SDN scheduler features, the status of each node can be prefetched to the SDN scheduler. Then, when a new job arrives, the scheduler divides the TK evenly, without creating a replica to the node, and assigns the tasks one by one to the nodes with the shortest completion time. In Figure 8, the movement time of TK1 is not required; the overall completion time of the prefetch method is 34 s, and TK8 becomes the last task to be completed among all nine tasks. Figure 9 shows the performance of different methods in terms of completion time. In the figure, the Pre-BASS scheme has the shortest completion time. However, in the Pre-BASS method, each node still completes the task at a different time.

*2.5. Research Gap*

We summarize the research gap as follows:

- In previous studies [5,6,17], the processing time of each task is the same by default. However, in reality, each node's performance can differ in multiple ways, that is, different hardware and processors are used for different nodes. In this study, we considered the different computation times of tasks on nodes.
- Previous studies [5,6,17] focus on task assignment and scheduling mechanisms. This study highlights the advantages of splitting tasks and rescheduling mechanisms, thereby improving network performance.

- This study also confirms that the use of a prefetching mechanism can improve performance compared with a non-prefetching mechanism.

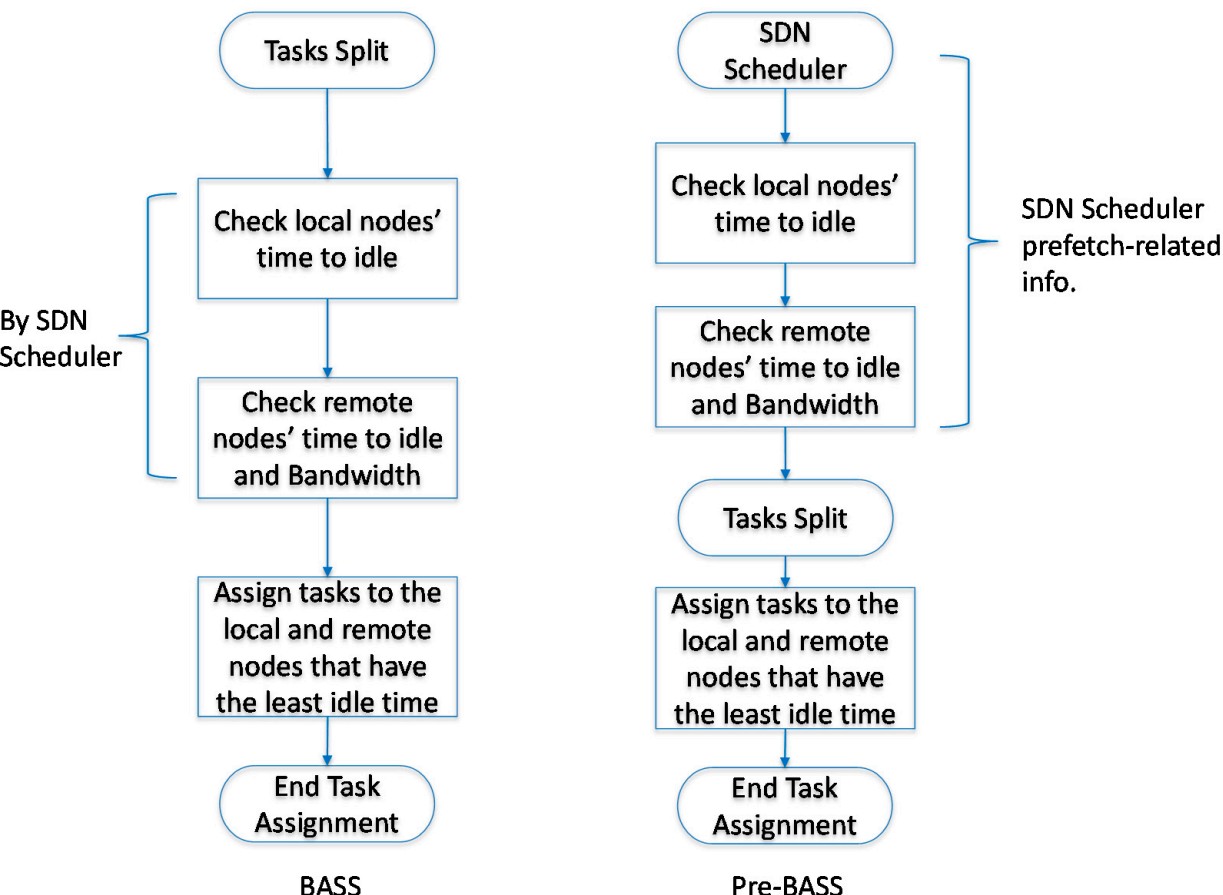

**Figure 6.** Flowchart of BASS (bandwidth-aware scheduling) and Pre-BASS.

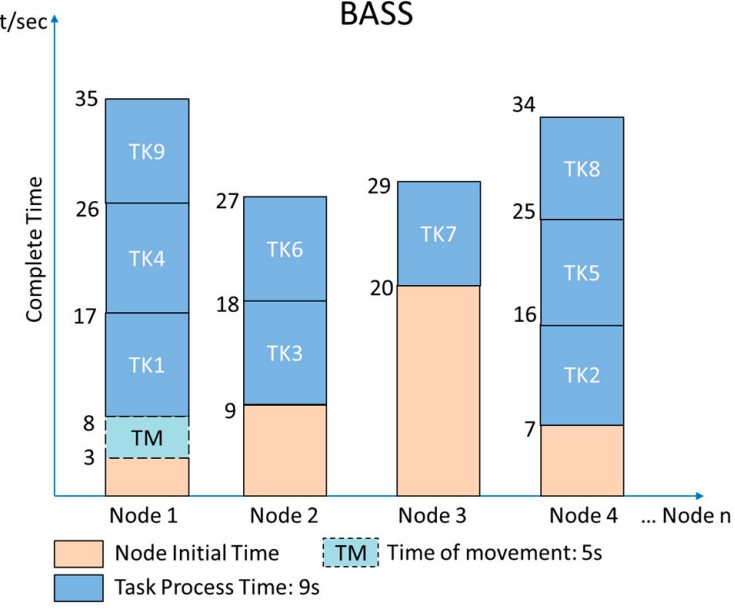

**Figure 7.** Example of BASS task assignments.

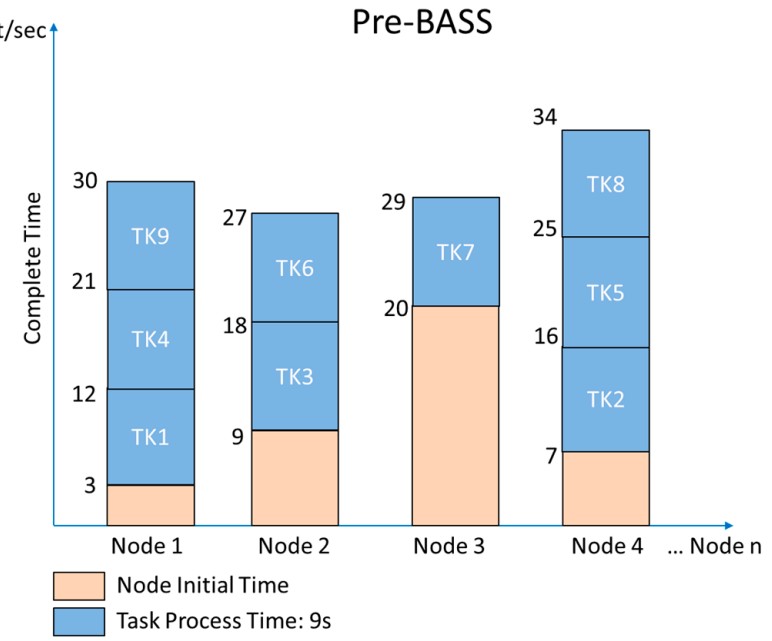

**Figure 8.** Example of Pre-BASS task assignments.

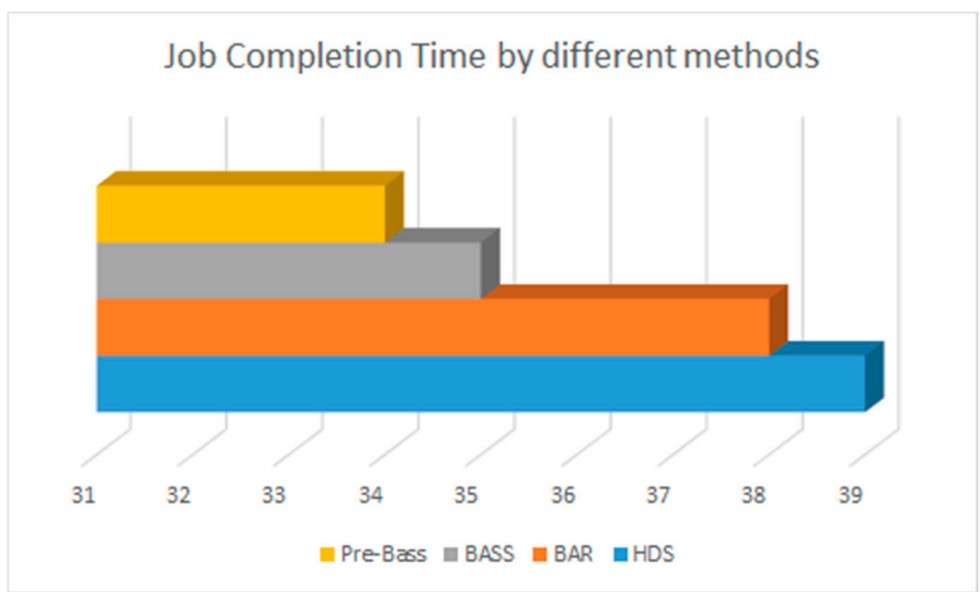

**Figure 9.** Job completion time in different methods.

## 3. Problem Formalization

This section describes problem formalization. Table 1 shows the notations used in the study: $TK_i$ denotes a task $i$ within a Hadoop job, $ND_j$ denotes a node $j$ in a Hadoop data center, $SZ_i$ denotes the size of the input split data for $TK_i$ assigned to $ND_j$; $TM_{i,j}$ denotes the data movement time from $ND_i$ to $ND_j$; $TP_{i,j}$ denotes the computation time of task $i$ on node $j$; $TE_{i,j}$ denotes the execution time of task $i$ on node $j$; $TS_i$ denotes the segmentation time of task $I$; $NP_j$ denotes the processing capacity of node $j$; $Idle_j$ denotes the time when $ND_j$ becomes idle; $C_{i,j}$ denotes the completion time of $TK_i$; $BW_{j,k}$ denotes the bandwidth between $ND_j$ and $ND_k$; and $BW_{ab}$ denotes the available bandwidth of a link. Based on the above notations, we obtain the following equations.

$$TM_{i,j} = SZ_i / BW_{i,j}, \tag{1}$$

$$TP_{i,j} = SZ_i/NP_j, \tag{2}$$

$$TE_{i,j} = TM_{i,j} + TP_{i,j}, \tag{3}$$

$$C_{i,j} = TE_{i,j} + Idle_j. \tag{4}$$

**Table 1.** Notations.

| Notations | Descriptions |
|---|---|
| $TK_i$ | A task $i$ within a Hadoop job |
| $ND_j$ | A node $j$ in the Hadoop data center |
| $SZ_i$ | Size of input split data for $TK_i$ assigned to $ND_j$ |
| $TM_{i,j}$ | Data movement time from $ND_i$ to $ND_j$ |
| $TP_{i,j}$ | The computation time of task $i$ on node $j$ |
| $TE_{i,j}$ | The execution time of task $i$ on node $j$ |
| $TS_i$ | Segmentation time of task $i$ |
| $NP_j$ | Process capacity of node $j$ |
| $Idle_j$ | The time when $ND_j$ becomes idle |
| $C_{i,j}$ | The completion time of $TK_i$ |
| $BW_{j,k}$ | Bandwidth between $ND_j$ and $ND_k$ |
| $BW_{ab}$ | The available bandwidth of a link |

Previous works do not account for the computational capacity of each node, assuming that the nodes have the same computation capacity. In this study, we consider a more realistic situation and discuss this part in detail. In Equation (2), the computation time of task $i$ on node $j$ (i.e., $TP_{i,j}$) is obtained in order that $SZ_i$ divided by $NP_j$ is $TP_{i,j}$. Equation (2) also denotes that each node has a different computational capacity, resulting in each node having a different $TP_{i,j}$. In Equation (5), the objective function is to find an available node that can produce the shortest completion time among all $n$ nodes of the data center:

$$ND_j = \text{argmin}_j\, C_{i,j}, \tag{5}$$

where $1 \leq j \leq n$. Equation (6) shows the SDN scheduler finding the slowest map or reducing task $TK_i$ to minimize the completion time of an entire job:

$$\min\{C_{i\prime,j\prime} = \max C_{i,j}(1 \leq i, i\prime \leq m, 1 \leq j, j\prime \leq n)\}, \tag{6}$$

where $m$ represents the task number of a job and $n$ is the node number of the Hadoop data center.

## 4. Proposed Scheme

### 4.1. Bandwidth-Aware Rescheduling

This section describes the proposed scheme BARE for SDN-based data centers. BARE includes four parts: global information collection, the task assignment scheme, the task splitting scheme, and the task rescheduling algorithm. Figure 10 shows the proposed BARE architecture. The dotted line represents control signaling, and the solid line represents data flow. One job can be divided into many small tasks. The scheduler module is responsible for scheduling tasks. If a task has a different priority, it is put in different queues. The number of block replicas is set to three, which is the default setting in Hadoop. Since most cloud computing systems are implemented on virtual hardware [22,23], data transfer cost significantly influences system performance [24,25].

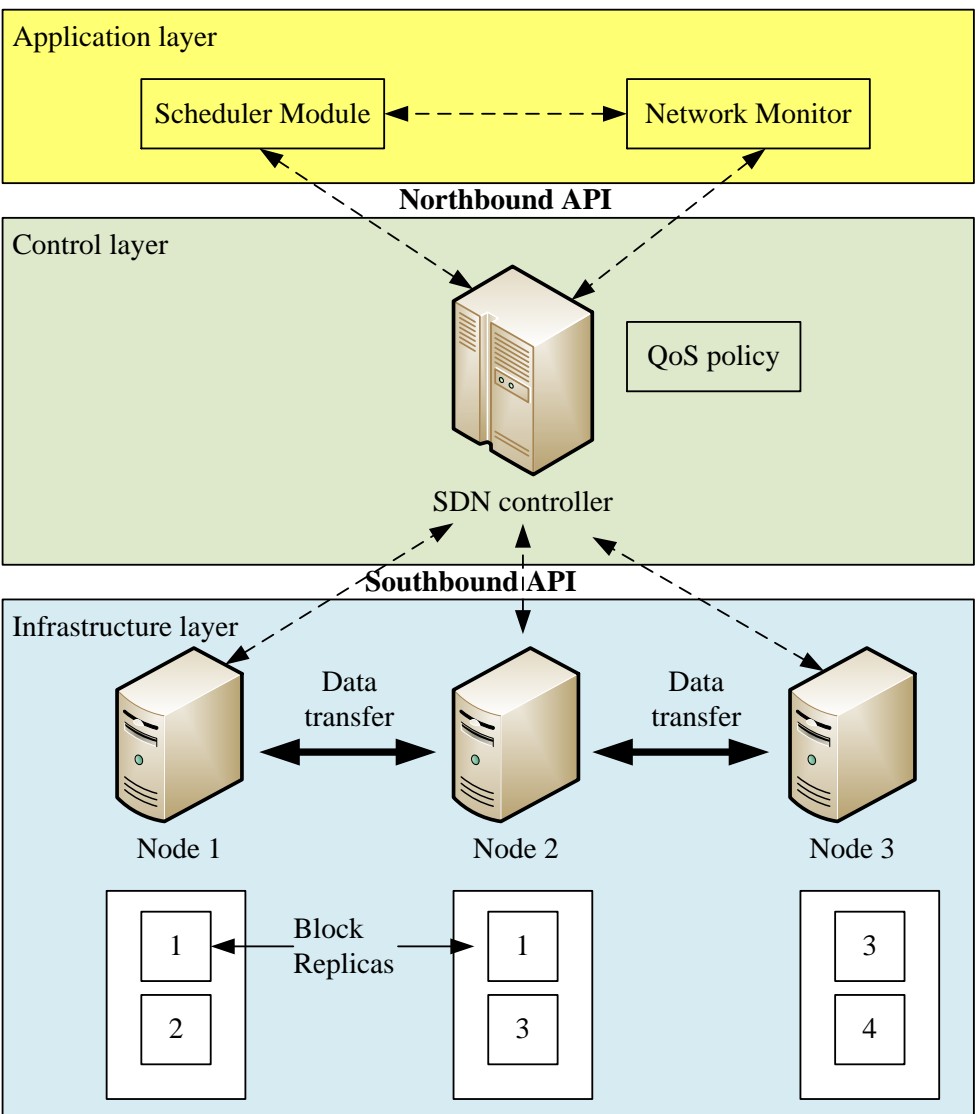

**Figure 10.** BARE architecture.

### 4.2. Global Information Collection

This section introduces the global information collection behavior of the centralized network monitor. In this study, we assume that each node periodically updates its information, and new job requests are held in the waiting queue. Node information includes the computational capacity of the node, the bandwidth between node $i$ and node $j$, the expected idle time, and the current workload. The network monitor then passes the information to the scheduler module. In addition, the scheduler module manages new job requests according to the scheduling policy. When the scheduler module obtains sufficient information, it can compute the value of $TM_{i,j}$, $TP_{i,j}$, $TE_{i,j}$, and $C_{i,j}$ through Equations (1)–(4).

### 4.3. Task Assignment Scheme

The main scheduling challenge in MapReduce is the requirement to place the computation close to data. Therefore, we hope that data are computed at the local node to increase throughput. Figure 11 shows the Hadoop MapReduce architecture. After selecting a job, the scheduler picks the map task with data closest to the slave. The priority order of task assignment is as follows. Node locality is ranked above rack locality, and rack locality is ranked above a remote node. The scheduler chooses another rack or other remote nodes to compute the data when the local node is overloaded. In the task assignment phase, our

BARE scheme is similar to the Pre-BASS scheme. BARE also uses the prefetching scheme to guarantee that the allocated tasks are optimal in terms of completion time. Then, the scheduler checks each remote task and allows its input split data to be prefetched/transferred before the available idle time, as early as possible. The final decision depends on the real-time residue bandwidth and computational capacity of the node. Note that when BARE prefetches a data block, it always moves the block starting with the least loaded node storing the replica to minimize the impact on overall performance.

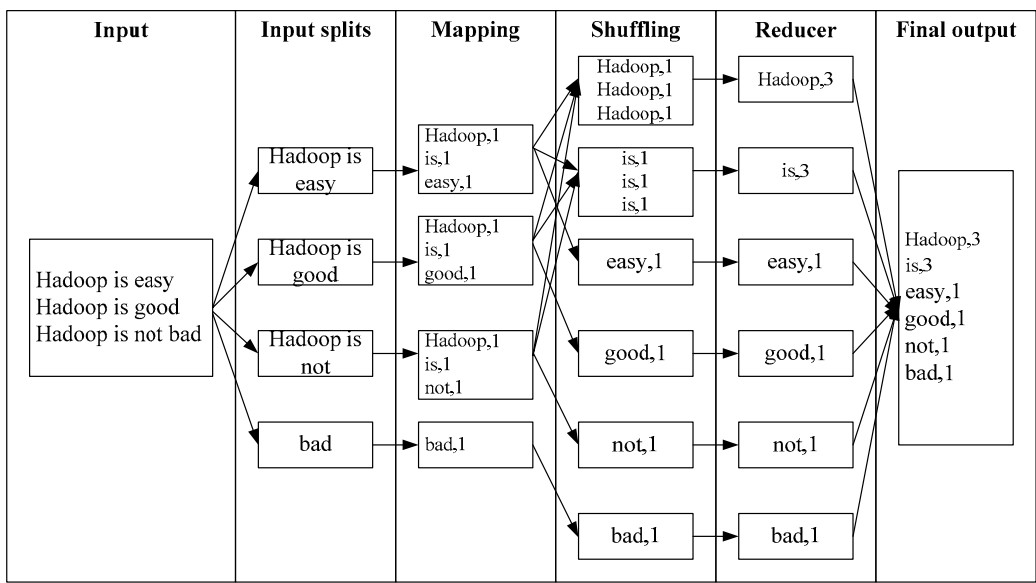

**Figure 11.** Hadoop MapReduce architecture.

Figures 3, 5, 7 and 8 show that Pre-BASS has a better result because it uses the prefetching scheme. Moreover, the scheduling algorithm needs to reduce the movement time of the task (i.e., $TM_{ij}$). Algorithm 1 shows the task assignment algorithm of BARE. The main differences between BARE and Pre-BASS are their detailed consideration of $TP_{i,j}$, task splitting, and rescheduling procedure. The computational capacity of each node is different. However, the BASS scheme assumes that each node has the same computational capacity. Therefore, we only modified this part in the task assignment procedure.

*4.4. BARE Rescheduling*

In this study, we propose an improvement scheme for reducing the completion time in SDNs. The BASS method divides every job equally into multiple tasks using the SDN scheduler. In previous studies [5,6,17], the processing capacity of every node is the same by default. However, in reality, a node's performance can differ in multiple ways; that is, different hardware and processors are used in different nodes, and many users use the bandwidth at the same nodes. The above factors influence the length of time required to complete a single task. It is important to understand how to manage the TP of each task for each node while also improving the completion time. Based on the FIFO HDS structure, the last two tasks are factors that influence the overall completion time. Referring to the task workflows in Section 2 for the four models, TK9 is the last task to complete, and only the Pre-BASS model has TK8 as the last task to complete. Therefore, we chose to focus on the last two tasks to reduce overall completion time. Concerning the split ratio, the SDN controller's decision is based on the TP of each node. Figure 12 shows an example of BARE. To easily present the advantages of BARE rescheduling, we assume that the TP of each node is the same (i.e., TP is 9 s). In the same environment, our scheme has the best result. In the figure, we split the last two tasks into four subtasks:TK8-1, TK8-2, TK9-1, and TK9-2. The subtasks are then moved to nodes with the least completion time to reduce the overall completion time of a job. In this case, the TCT of BARE is 32 s. It can be inferred that the

advantages of BARE are more obvious if we consider the different TPs of each node. The following algorithm is added to the original BASS algorithm, as shown in Algorithm 2. Table 2 describes the notations.

---

**Algorithm 1: BARE algorithm**

---

INPUT:
Given the submitted job with *m* tasks $TK_i$, $Idle_j$, $TP_{ij}$, data size $SZ_i$, and the *n* nodes $ND_j$ in a Hadoop cluster.
EXECUTE:
   SDN controller performs the prefetching scheme
FOR (*i*=1 to m)
   Application layer module starts to:

(1)    Network monitor collects global information and talks to the scheduler module
(2)    Scheduler module computes the idle time
(3)    Assign a task to the node (priority: local node > rack node > remote node)

      •   $TE_{new} > TM_{i,j} + TP_{new} \rightarrow$ without change
      •   $TE_{new} = TM_{i,j} + TP_{new} \rightarrow$ without change
      •   $TE_{new} < TM_{i,j} + TP_{new} \rightarrow$ change

   IF (task completion time of each node < threshold)
     Without change
   ELSE

(1)    Perform the task splitting
(2)    Rescheduling

   End IF
End FOR
Return the assignment results for all *m* tasks.

---

**Algorithm 2: Task splitting and rescheduling algorithm**

---

With job scheduler task assignment to split last 2 tasks:
      $TK_{m-1} = STK_{m-1\_1} + STK_{m-1\_2}$
$TK_{m-2} = STK_{m-2\_1} + STK_{m-2\_2}$
for (z= 1, 2) do
      Find $ND_{loc}$ with available idle time $Idle_{loc}$ for $STK_{m-z\_2}$
Find $ND_{minnow}$ with available idle time $Idle_{minnow}$ for $STK_{m-z\_2}$
If ($ND_{minnow}$ != $ND_{loc}$ && $Idle_{loc} > Idle_{minnow} + (1-SP_{m-z})TM + (1-SP_{m-z})TP$)
Assign $STK_{m-z\_2}$ to $ND_{loc} \equiv ND_{minnow}$
end if
end for

---

In this case, the difference in TP is solved because the last two tasks are evenly split into four subtasks in BARE. Then, an evaluation for the node with the least completion time, the subtask movement time, and the processing time is performed. After the evaluation, if a new node with the second portion of the subtask can complete the task faster than the original node, the second portion of the subtask is transferred to the new node. This evaluation is performed twice for the last two tasks. Note that the new evaluation time $C_{i,j}$ includes $(1 - SP_{m-z})TM_{i,j} + (1 - SP_{m-z})TP_{i,j} + Idle_{minnow}$.

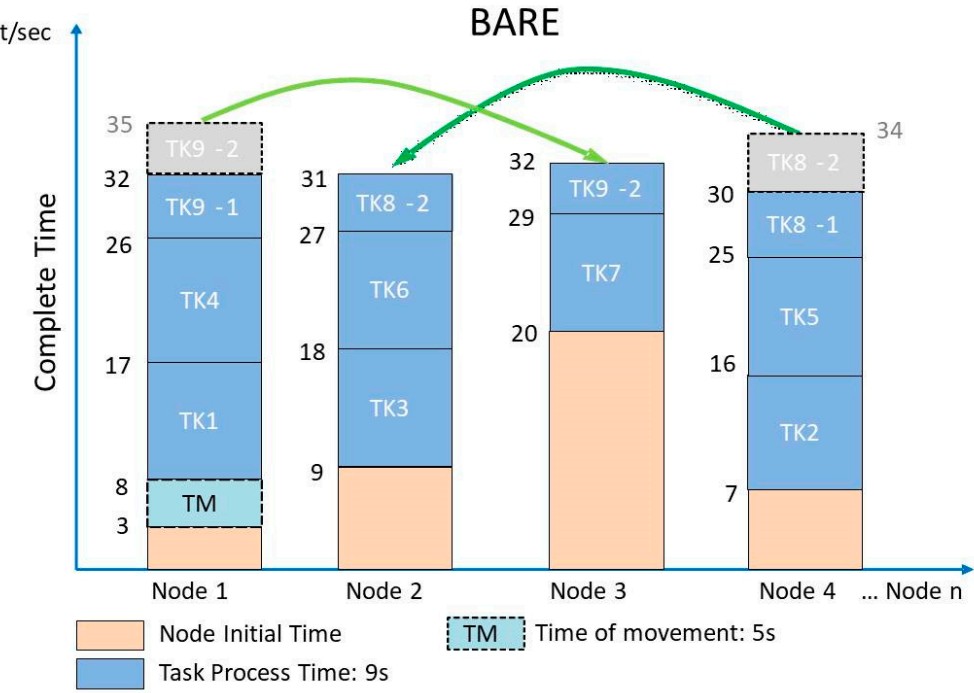

**Figure 12.** Example of BARE (bandwidth-aware rescheduling).

**Table 2.** Notations.

| Variables | Meaning |
|---|---|
| $TK_{m-z}$, z = 1, 2 | The last 1 or 2 tasks among the split job |
| $SP_{m-z}$, z = 1, 2 | Split ratio, $0 \leqq SP_{m-z} \leqq 1$ |
| $STK_{m-z\_1}$ | The first portion of the subtask from $TK_{m-z}$, TP = $SP_{m-z}$% |
| $STK_{m-z\_2}$ | The second portion of the subtask from $TK_{m-z}$, TP = $(1 - SP_{m-z})$% |

## 5. Performance Evaluation

This study considers two performance metrics: the TCT and RDL. We compared the BARE scheme with the HDS, BAR, BASS, and Pre-BASS schemes. Table 3 shows the simulation parameters. We set the number of tasks from 10–100, the number of node ranges from 10–100, and the initial workload of each node from 10%–80%. The bandwidth between nodes was 100–1000 Mb/s, the data size was 100–1000 MB, and the number of block replicas was three. Each simulation test was run 10 times, and we used the average value for comparison.

1.　TCT: A shorter TCT means better system performance. We hope the request of the user can be completed as fast as possible.
2.　RDL: The priority order of task assignment is as follows. Node locality is ranked above rack locality, and rack locality is ranked above a remote node; this is because the performance of node locality is the best. If the RDL is high, it also means the scheduling algorithm is better. Equation (7) shows the RDL.

$$RDL = \frac{numbers\ of\ data\ locality}{numbers\ of\ total\ data}. \tag{7}$$

**Table 3.** Simulation parameters.

| Parameters | Values |
|---|---|
| Number of tasks | 10–100 |
| Number of nodes | 10–100 |
| The initial workload of each node | 10%–80% |
| Bandwidth between nodes | 100–10,000 Mb/s |
| Data size | 100–1000 MB |
| Block replicas | 3 |

Figures 13–15 show the TCT performance with different parameters, including bandwidth, data size, and the number of nodes. Figure 13 shows that BARE has the best result. The HDS does not use the bandwidth as a parameter for task allocation. BAR is based on the allocation of the HDS and attempts to adjust the latest task completed on a job. If the latest task completed in other nodes can be completed in advance, the task is redistributed to reduce the job completion time. BASS also uses network bandwidth as a parameter for task assignment. Before task assignment, however, the maximum bandwidth in a network needs to be sliced by time. Compared with BARE, this is an additional task, and BARE makes full use of a network's maximum capacity. The remaining bandwidth is more suitable for actual scenarios. In addition, when the bandwidth is sufficiently large, the data movement time (i.e., $TM_{ij}$) from node $i$ to node $j$ approaches zero. Therefore, the TCT of all methods will be drastically reduced. Figure 14 shows a trend that the TCT decreases when nodes are added; this is because more nodes can reduce the amount of workload. According to the results, BARE significantly reduces the TCT using task splitting and rescheduling mechanisms. Note that more nodes signify more cost. Performance and cost are a trade-off problem. This issue is beyond the scope of our study. Figure 15 shows that the bigger the data size, the longer the TCT. BARE is proven to have the best results. In a large data size environment (i.e., data size is 1000 MB), BARE improves the performance of the HDS, BAR, BASS, and Pre-BASS by 45%, 31%, 12%, and 10%, respectively. The task splitting and rescheduling mechanisms can effectively shorten the TCT.

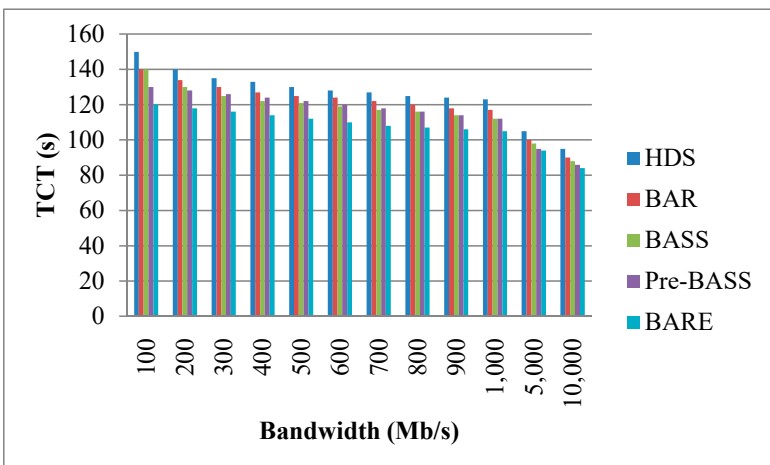

**Figure 13.** Effect of bandwidth on TCT (number of nodes is 30). TCT: task completion time.

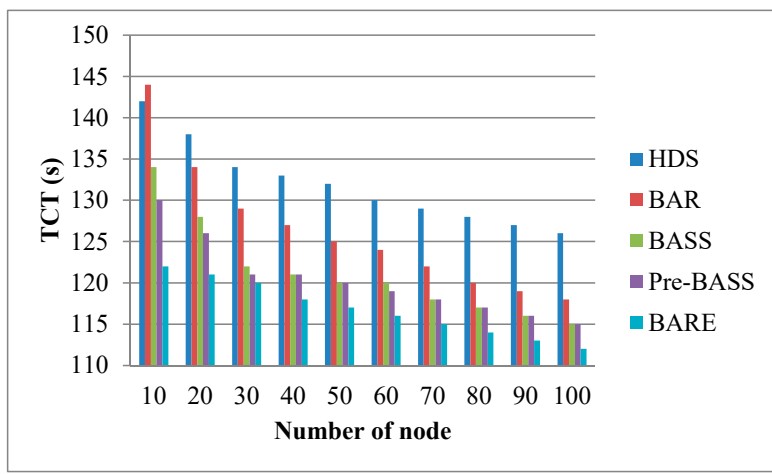

**Figure 14.** Effect of node number on TCT (bandwidth is 400 MB/s).

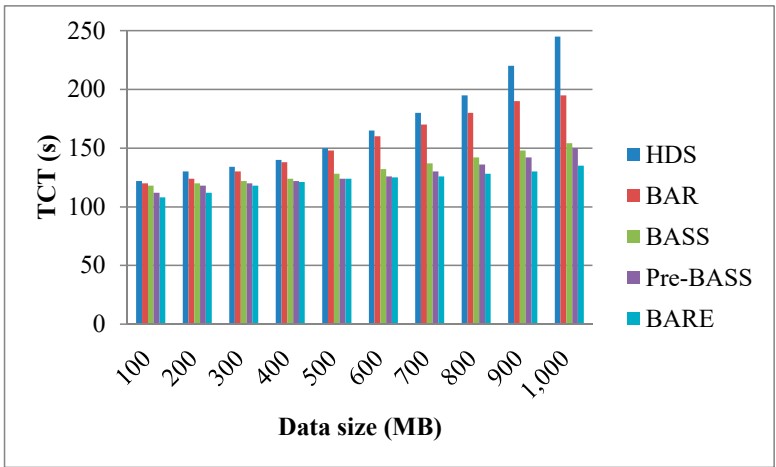

**Figure 15.** Effect of data size on TCT (bandwidth is 400 MB/s).

Figures 16–18 show the performance of the RDL with different parameters. Typically, the processing time of a task is short when data are allocated to the local node. Therefore, the RDL is a key factor for scheduling algorithms in data centers. The RDL of BARE and Pre-BASS is high because they use a prefetching mechanism in the task assignment phase. The prefetching scheme guarantees that the allocated tasks are all optimal in terms of completion time. Generally, the TCT reduces when tasks are assigned to the same local node. Figure 16 shows that the bandwidth of the link increases (i.e., signifying that $TM_{i,j}$ decreases), and the RDL is slightly decreased. When the movement time and processing time of other rocks are less than the processing time of the local rack, the data are not processed locally and are then transferred to other racks (i.e., it means the RDL decreases). Interestingly, we can observe a specific situation taking place. When the bandwidth is sufficiently large, the overall execution time and RDL are reduced; this is because the data movement time is close to zero. Therefore, when the execution time of the remote node, along with the data movement time, is less than the execution time of the local node, the central scheduler will select the remote node to perform the work. In Figure 17, the number of nodes increases, and the RDL decreases. In Figure 18, the data size increases, and the RDL also increases. These results emphasize that our BARE scheme works best. BARE collects global information, uses the prefetching scheme to improve the performance of the local node, optimally assigns tasks to the most suitable node, and performs the rescheduling procedure.

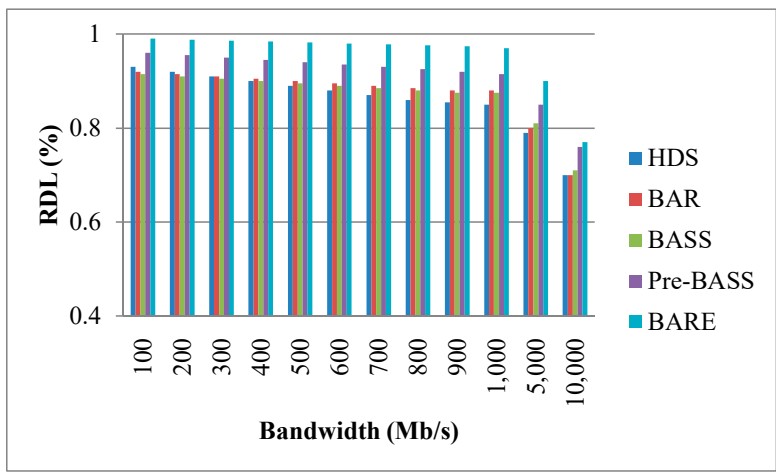

**Figure 16.** Effect of bandwidth on RDL (ratio of data locality).

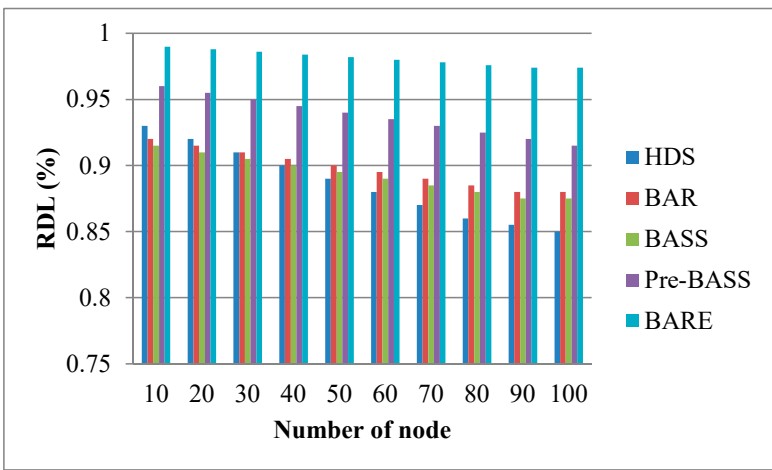

**Figure 17.** Effect of node number on RDL.

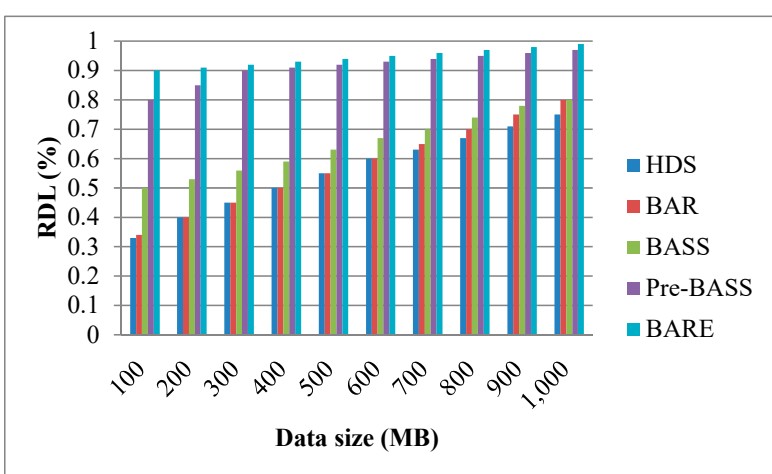

**Figure 18.** Effect of data size on RDL.

## 6. Conclusions

In reality, data centers handle a greater amount of work and more complex situations. In addition, achieving minimum TCT in the Hadoop system is an NP-complete problem. Therefore, data center networks need an efficient job scheduling scheme to improve network performance. In this study, we propose a feasible solution in an SDN-based data

center network, called the BARE mechanism. In BARE, the network monitor collects global information and communicates with the scheduler module. The scheduler module organizes data processing, executes task prefetching and allocation plans, executes task splitting methods, and implements rescheduling mechanisms to reduce the TCT. According to the simulation results, the proposed scheme outperforms other existing schemes by adopting task splitting and rescheduling schemes. In terms of TCT, BARE outperforms the HDS, BAR, BASS, and Pre-BASS schemes by approximately 25%, 17%, 10%, and 7%, respectively. In terms of RDL, when the data size is small (i.e., 100 MB), BARE outperforms the HDS, BAR, BASS, and Pre-BASS schemes by approximately 58%, 56%, 40%, and 10%, respectively.

Energy is also an important issue in data center networks. However, performance and energy (or cost) are trade-offs. Therefore, we will consider the issue of energy consumption in our future work. In addition, the centralized scheduler may become a performance bottleneck. Thus, we will attempt to propose a distributed solution.

**Author Contributions:** Conceptualization, M.-C.C.; methodology, M.-C.C., C.-C.Y. and C.-J.H.; software, C.-C.Y. and C.-J.H.; investigation, M.-C.C.; resources, M.-C.C.; data curation, C.-C.Y. and C.-J.H.; writing—original draft preparation, M.-C.C. and C.-J.H.; writing—review and editing, M.-C.C., C.-C.Y. and C.-J.H.; supervision, M.-C.C. All authors have read and agreed to the published version of the manuscript.

**Funding:** This research was funded by the Ministry of Science and Technology, R.O.C., under grants MOST 107-2221-E-163-001-MY3.

**Data Availability Statement:** The funders had no role in the design of the study; in the collection, analyses, or interpretation of data; in the writing of the manuscript, or in the decision to publish the results.

**Conflicts of Interest:** The authors declare no conflict of interest.

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
