# Peer review of "Bandwidth-Aware Rescheduling Mechanism in SDN-Based Data Center Networks"

_electronics, doi:10.3390/electronics10151774_

Round 1

Reviewer 1 Report

This article suffers from a great deal of misunderstanding and confusion about the architecture of SDN and its relationship with the cloud. It might have been an acceptable presentation of ideas, if it had dropped any reference to SDN and 5G, and simply focused on rescheduling mechanisms in the cloud. The ideas presented in this paper have nothing to do with SDN.

The paper presents its architecture in Figure 11, which indicates a major misunderstanding of SDN architecture.  The SDN controller is responsible for communicating and controlling network switches, not the computing servers. What you have in mind is actually a central scheduler, that might be running on the same server as an SDN controller, but it certainly is NOT an SDN controller because it is not controlling network switches. SDN controller has nothing to do with scheduling application-layer jobs.

The paper needs a major revision. In doing so, the following problems must be addressed:

1. The abstract is had to read and suffers from poor English. Please write it from scratch with a clearer focus on describing the actual work and results of the paper.

2. The introduction section is quite out of focus and very confusing. The proposed work has nothing to do with 5G or IoT; those paragraphs must be dropped from the paper. The confusion regarding the role of SDN controller must be addressed. SDN controller and job scheduler are not the same entities even if they may be co-located.

3. The entire notion of this work being "SDN-based" must be dropped; in fact, I recommend removing any mention of SDN in this paper. This work should be more accurately described in cloud computing terms. While Cloud data centers today often employ SDN, SDN is not a requirement for cloud or vice versa. Just collecting global network data does not make the network an SDN, unless you use those data to make specific network-layer flow modifications, which is not the case here.

4. The English language and writing style of the paper needs improvement, specially in the abstract and first two sections.

Author Response

We would like to thank the reviewers for their valuable suggestions on the manuscript. All comments have been read thoroughly and the paper is revised accordingly. The point-by-point responses to each reviewer comment are listed as follows. The comments are in italic and the modified texts in the revised manuscript are highlighted.

Reviewer 2 Report

This study aims to finding an efficient task scheduling method for reducing the task completion time. In general, this study is well purposed and designed. However, there are some areas that should be improved.

  1. In introduction, it is not common that the introduction contains figures and tables. The introduction should focus on the rationals of the study and gaps in the literature as well as the main purpose of the study. I see some of these contents, but not all, particularly the gaps in existing literature.
  2. I see some expression in future. For example, line 146, "This section will ...." The research should be done, not for the future.
  3. In 5. Problem evaluation section, authors(s) should discuss the results more.
  4. The conclusion section should be revised with a great attention by including the summary of findings, academic and practical implications, and limitation with future study direction.

Author Response

(The authors gave the same response as above.)

Round 2

Reviewer 1 Report

I would like to thank the authors for taking my previous feedback into account. The paper has been notably improved. Some further revisions would make it ready for publication:

1. Related work: I suggest adding a section titled "Research Gap" that clearly spells out what is missing in prior work, and which parts of the research gap are addressed by this paper. 

2. Section 2.3: the statement that the BAR scheme [5] "leverages Hadoop and SDN-based network" seems to be inaccurate because the reference [5] does not mention SDN at all. I suggest replacing "SDN-based network" here with "centralized scheduler". Also given that Ref[5] is fairly old (2011), I suggest checking subsequent works based on BAR and including them in the literature review.

3. Section 2.2: In the section title, add the abbreviation you have used later (HDS).

4. Line 167: by "earliest" I assume you mean "shortest"? These two terms are different and may cause confusion.

5. line 169: I suggest using "scheduler" instead of "Controller" here.

6. Section 4.4: The BARE task processing time is set to 9 seconds for all tasks. Is this number hard coded into your scheme, or simply a simulation parameter? Please provide justification for this value. If this is merely a simulation parameter, please move it to section 5.

7. Section 5: Line 267 states the initial workload is randomly generated. What sort of random distribution? is it uniform? please provide a mathematically-accurate description and justify it.

8. Table 3- Simulation parameters: Link bandwidths are set to 100-1000 Mbps, which seems fairly low for a data center. Please include additional simulations at higher rates (e.g. 10-100 Gbps) to see how the reduced TM would affect the results. 

9. Overall, the paper requires significant editing for English language and grammar. please consider using a professional service for proofreading it. 

Reviewer 2 Report

Author(s) addressed most of the previous comments very well. However, as pointed out in the previous review, the conclusion section should be extended more with implications from academic and practical perspectives. The conclusion section has not been revised as I requested as followed. 

  1. The conclusion section should be revised with great attention by including the summary of findings, academic and practical implications, and limitations with future study direction.

Round 3

Reviewer 2 Report

I am now fine with all revision. Thank you for hard works.